# Not All Inputs Are Valid: Towards Open-Set Video Moment Retrieval Using Language

## ABSTRACT

As a significant yet challenging multimedia task, Video Moment Retrieval (VMR) targets to retrieve the specific moment corresponding to a sentence query from an untrimmed video. Although recent respectable works have made remarkable progress in this task, they implicitly are rooted in the closed-set assumption that all the given queries as video-relevant[1]. Given a video-irrelevant OOD query in open-set scenarios, they still utilize it for wrong retrieval, which might lead to irrecoverable losses in high-risk scenarios, *e.g.*, criminal activity detection. To this end, we creatively explore a brand-new VMR setting termed Open-Set Video Moment Retrieval (OS-VMR), where we should not only retrieve the precise moments based on ID query, but also reject OOD queries. In this paper, we make the first attempt to step toward OS-VMR and propose a novel model **OpenVMR**, which first distinguishes ID and OOD queries based on the normalizing flow technology, and then conducts moment retrieval based on ID queries. Specifically, we first learn the ID distribution by constructing a normalizing flow, and assume the ID query distribution obeys the multi-variate Gaussian distribution. Then, we introduce an uncertainty score to search the ID-OOD separating boundary. After that, we refine the ID-OOD boundary by pulling together ID query features. Besides, video-query matching and frame-query matching are designed for coarse-grained and fine-grained cross-modal interaction, respectively. Finally, a positive-unlabeled learning module is introduced for moment retrieval. Experimental results on three challenging datasets demonstrate the effectiveness of our OpenVMR. Codes will be released upon acceptance.

## CCS CONCEPTS

• **Information systems → Video search**.

## KEYWORDS

Open-set Video Moment Retrieval, ID Query, OOD Query

## 1 INTRODUCTION

Video Moment Retrieval (VMR) is a challenging yet crucial task in multi-modal retrieval [8, 34, 66, 71, 82], which has attracted significant attention in recent years due to its vast potential applications

---

[1]In this paper, we treat "video-relevant query" as "in-distribution (ID) query" and "video-irrelevant query" as "out-of-distribution (OOD) query".

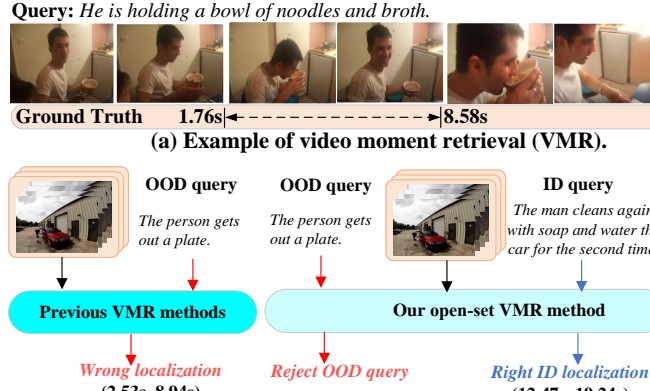

**Query:** *He is holding a bowl of noodles and broth.*

Ground Truth      1.76s ◄------------► 8.58s

**(a) Example of video moment retrieval (VMR).**

OOD query — *The person gets out a plate.* — OOD query — *The person gets out a plate.* — ID query — *The man cleans again with soap and water the car for the second time.*

Previous VMR methods — Our open-set VMR method

*Wrong localization* (2.53s, 8.94s) — *Reject OOD query* — *Right ID localization* (12.47s, 19.24s)

**(b) Comparison between previous methods and our method.**

**Figure 1: (a) Example of the video moment retrieval (VMR) task. (b) Comparison between previous VMR models and our open-set VMR model. Given a video and a query, previous methods directly conduct retrieval, regardless of whether the query is video-relevant (ID) or video-irrelevant (OOD). Our model can reject OOD queries and recognize ID queries for moment retrieval.**

in areas such as human activity retrieval [21, 54, 77, 84]. As illustrated in Fig. 1(a), its main objective is to identify and retrieve the relevant video moment corresponding to a given sentence query. Obviously, most of the video content is query-irrelevant, and only a very short video segment matches the query. It is substantially more challenging since a well-designed model should not only model the complex multi-modal interaction between videos and queries, but also capture complicated context information for cross-modal semantics alignment. The target model requires recognizing objects/activities and identifying which visual content is sufficient to retrieve the accurate moment expressed in free-form natural language, accounting for the fact that the accurate moment may occupy only a tiny portion of the entire video. To achieve this, both videos and queries must be deeply integrated to distinguish the subtle details of adjacent frames, thereby enabling accurate determination of moment boundaries.

Most existing VMR works [5, 15, 35, 72, 73, 76, 79] are under fully-supervised setting, where each frame is manually labeled as the query-relevant or query-irrelevant frame. Instead of using such dense frame annotations, some recent works try to explore a weakly-supervised setting [6, 33, 40, 55, 83] with only the video-query correspondence to alleviate the reliance on a certain extent. However, their performance is less satisfactory. Although the above VMR methods have made exciting headway, they refer to the close-set assumption that we can obtain a moment in the untrimmed video for any given query. In the real-world open-set environment, we often input a random or irrelevant query for moment retrieval. As shown in Fig. 1(b), given an irrelevant query, previous methods still retrieve a wrong moment as the model output, which will lead to irrecoverable losses in high-risk scenarios, *e.g.*, criminal activity

detection. It is unacceptable for our society to classify normal activity as criminal and to treat criminal activity as normal. In real-world multimedia applications, we have a small-scale set of video-query pairs, and many unannotated videos and many video-irrelevant queries. Since labeling video-query annotation is very expensive and time-consuming, it is unrealistic to manually annotate all the queries as video-relevant (*i.e.*, ID) or video-irrelevant (*i.e.*, OOD).

Therefore, we explore a novel and challenging task: open-set VMR (OS-VMR). Given an untrimmed video, our OS-VMR task aims to not only temporally retrieve the specific moment semantically corresponding to the ID query, but also reject the OOD query shown in Fig. 1(b). Different from previous closed-set settings that only align video and query representations for moment retrieval, our OS-VMR task suffers from three major challenges: 1) *how to accurately learn the distribution of ID queries?* 2) *how to precisely reason the separating boundary of ID and OOD queries?* 3) *how to fully interact with video and ID queries?* In this paper, we propose a novel OpenVMR framework for the challenging OS-VMR task. Specifically, we first design a multi-layer coupling block to construct the normalizing flow for learning ID query distribution based on the multi-variate Gaussian distribution assumption. Besides, we reason the ID-OOD separating boundary by a well-designed uncertainty score and the log-likelihood distribution of each query. Moreover, we pull ID query features together to refine the ID-OOD boundary based on a triplet loss for OOD query detection. After that, for the video and ID query, we conduct the cross-modal interaction for video-query matching and frame-query matching. Finally, we design a simple yet effective positive-unlabeled learning module with pre-defined proposals to retrieve the target moment.

Our main contributions are summarized as follows:

- To the best of our knowledge, we make the first attempt at the open-set Video Moment Retrieval (OS-VMR) task, which is fundamentally more challenging but highly valuable in open-set settings. In this setting, we should not only retrieve the video moment for ID queries, but also reject OOD queries.
- To address our challenging OS-VMR task, we propose a general OpenVMR framework that first distinguishes ID and OOD queries by the normalizing flow technology, and then utilizes ID queries for moment retrieval.
- We conduct extensive experiments on three popular VMR datasets (ActivityNet Captions, Charades-STA and TACoS). Experimental results on both open-set and closed-set settings show that our proposed model outperforms other state-of-the-art approaches by a large margin.

## 2 RELATED WORK

**Video moment retrieval.** The goal of Video Moment Retrieval (VMR) is to recognize and temporally localize all the action instances in an untrimmed video [5, 43, 79]. Most of the existing VMR methods [3, 12, 47, 72, 81, 82] refer to the fully-supervised setting where all video-query pairs are annotated in detail, including corresponding moment boundaries. The above methods heavily rely on datasets that require numerous manually labeled annotations for training. To ease the human labeling efforts, several recent works [6, 33, 55, 83] consider a weakly-supervised setting that only accesses the information of matched video-query pairs

without accurate moment boundaries. However, their performance is significantly worse than fully-supervised methods.

**Open-set recognition.** Open-set Recognition (OSR) aims to classify in-distribution (ID) samples and reject out-of-distribution (OOD) samples. Previous OSR methods [18, 28, 32, 36, 39, 41, 56, 60, 69, 70, 74] can be divided into four types: classification-based methods [17, 27, 29, 32], density-based methods [23, 52], distance-based methods [28, 57] and reconstruction-based methods [67, 86]. 1) Early OSR works refer to a classification framework, which utilizes the maximum softmax probability to determine the ID/OOD samples. 2) To more explicitly model ID, density-based OSR methods are proposed to leverage the probabilistic models for OSR. These methods are under an operating assumption that OOD samples have low likelihoods whereas ID samples have high likelihoods under the estimated density model. 3) The distance-based OSR methods are based on an intuitive idea that OOD samples should be relatively far away from the centroids of ID samples. 4) The reconstruction-based methods often leverage the encoder-decoder framework, which is trained on only ID samples and generates different outcomes for OSR. Inspired by OSR, we step further toward the OS-VMR problem in this paper. Please note that existing OSR methods focus on open-set object detection [9, 19, 20, 38, 45, 49, 61] in the image datasets. These OSR methods cannot effectively understand video and query in our VMR task. However, it is the uniqueness of the moment retrieval in an open-set setting that makes the OS-VMR problem even more challenging and valuable in practice.

## 3 OUR PROPOSED OPENVMR

**Setup.** Given an untrimmed video $V$ and corresponding language query $Q$, the traditional Video Moment Retrieval (VMR) task aims to retrieve the query-described activity moment from the video. However, existing VMR methods refer to a closed-set setting, which lacks their understanding of video-irrelevant queries, limiting their real-world information retrieval applications.

To this end, we investigate a more practical but challenging setting, called open-set VMR (OS-VMR), which not only conducts video grounding by the video-relevant query but also rejects the video-irrelevant query. Given an untrimmed video and a sentence query, the OS-VMR task aims to retrieve the moment location from the video (if the query is video-relevant, *i.e.*, ID query), or reject the query (if the query is video-irrelevant, *i.e.*, OOD query). Therefore, our posed OS-VMR is brand-new and more challenging than VMR. Fig. 2 illustrates the framework of our proposed method.

### 3.1 Preparation

**Video encoder.** Following previous VMR works [79, 81, 82], given a video with $N_v$ frames, we first extract its frame-wise features by a pre-trained C3D network [58], and then employ a multi-head self-attention [59] module to capture the long-range dependencies among video frames. We denote the extracted video features as $V = \{v_i\}_{i=1}^{N_v} \in \mathbb{R}^{N_v \times d}$, where $d$ is feature dimension.

**Query encoder.** Similarly, given a query with $N_w$ words, we also follow previous VMR works [79, 81, 82] to utilize the Glove [48] embedding to encode each word into dense vector. We further employ the Bi-GRU [7] layers to extract word-level query feature $W = \{w_j\}_{j=1}^{N_w} \in \mathbb{R}^{N_w \times d}$. To capture the global semantics of the

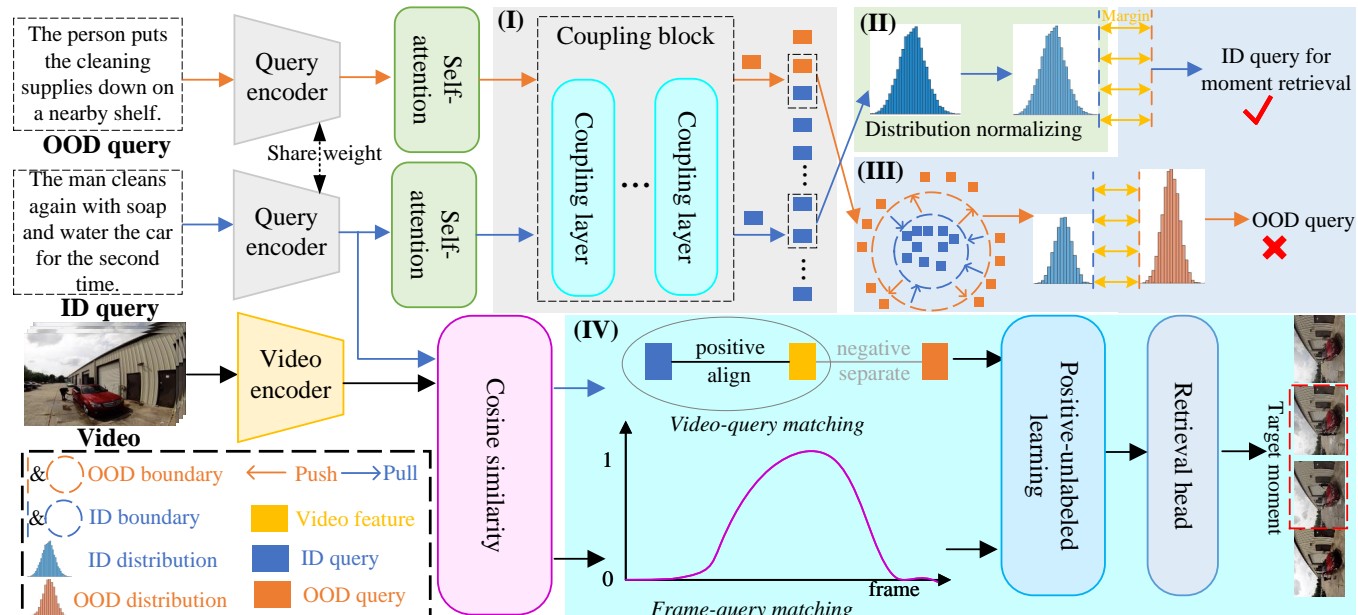

**Figure 2: Framework of our proposed method for open-set VMR, where module (I) is "ID knowledge acquisition", module (II) is "uncertainty-aware OOD boundary reasoning", module (III) is "ID-OOD boundary refinement", and module (IV) is "cross-modal interaction". Firstly, we employ video and query encoders to extract corresponding features. In module (I), we then design a coupling block based on the normalizing flow technology to learn ID query feature distribution as ID knowledge. Besides, we define the uncertainty score to reason the ID-OOD boundary in module (II). In module (III), we further refine the ID-OOD boundary by maximizing the margin between ID and OOD boundaries. Finally, we utilize the ID query for final moment retrieval by: a) we align video and ID query by video-query matching and frame-query matching, b) treating query-relevant frames as positive data and query-irrelevant frames as unlabeled data, we introduce positive-unlabeled learning to predict the coarse-grained moment boundary, and c) a retrieval head is used for fine-grained moment retrieval. Best viewed in color.**

whole query, we utilize the Skip-thought parser [24] to extract the sentence-level query feature $q \in \mathbb{R}^d$.

## 3.2 ID Knowledge Acquisition

When it comes to detecting OOD queries, we aim to find an OOD-independent separating boundary to distinguish ID queries and OOD queries. Thus, we extract the simplified distribution of ID query features. We use normalizing flow [10, 11, 26, 46, 68] to learn the distribution of ID query features. In a VMR dataset, for a video, we select $N_{id}$ ID queries and $N_{ood}$ OOD queries. We refer to the sentence-level query features extracted by the query encoder as input features for normalizing flow. We denote these features as $Q = Q^{id} \cup Q^{ood}$, where $Q^{id} = \{q_i^{id}\}_{i=1}^{N_{id}}$ and $Q^{ood} = \{q_j^{ood}\}_{j=1}^{N_{ood}}$ are ID and OOD query features, respectively.

*3.2.1 Normalizing Flow Construction.* Firstly, we denote $\Phi_\omega : Q \in \mathbb{R}^d \rightarrow \mathcal{X} \in \mathbb{R}^d$ as our normalizing flow, where $\omega$ is a learnable parameter and $\mathcal{X}$ denotes the latent space. Especially, a coupling block with multiple coupling layers [10] is leveraged such that $\Phi_\omega = \Phi_C \circ \cdots \circ \Phi_2 \circ \Phi_1$, where $C$ is the total layer number.

Defining $d$-dimensional input and output features of normalizing flow as $k_0 = q \in Q$ and $k_C = x \in \mathcal{X}$, the output of the $c$-th latent layer is $k_c = \Phi_c(k_{c-1})$, where $\{k_c\}_{c=1}^{C-1}$ are the intermediate outputs. By the change of variables formula, the input distribution estimated by model $p_\omega(q)$ can be formulated as:

$$\log p_\omega(q) = \sum_{c=1}^C \log |\det J_{\Phi_c}(k_{c-1})| + \log p_{\mathcal{X}}(\Phi_\omega(q)), \quad (1)$$

where $J_{\Phi_c}(k_{c-1}) = \partial \Phi_c(k_{c-1}) / \partial k_{c-1}$ is the Jacobian matrix of $\Phi_c$ at $k_{c-1}$, and det means determinant. Besides, we can approximate the feature distribution $p_Q$ with $p_\omega(q)$ by the normalizing flow.

By optimizing the log-likelihoods across the training distribution $p_Q$, we can obtain the set of parameters $\omega$ as follows:

$$\omega^* = \underset{\omega}{\arg\min} \, \mathbb{E}_{q \sim p_Q} [-\log p_\omega(q)]. \quad (2)$$

*3.2.2 Learning ID Query Feature Distribution.* Then, by maximum likelihood optimization, we leverage the normalizing flow to learn ID query feature distribution. In general, the latent variable distribution $p_{\mathcal{X}}(x), x \in \mathbb{R}^d$ can be assumed to obey the following multi-variate Gaussian distribution [16]:

$$p_{\mathcal{X}}(x) = (2\pi)^{-\frac{d}{2}} \det(\sigma^{-\frac{1}{2}}) \exp[-\frac{1}{2}(x-\mu)^\top \sigma^{-1}(x-\mu)], \quad (3)$$

where $\mu$ and $\sigma$ are the mean and the covariance, respectively. For simplicity, we assume the latent variables for the ID query feature to obey the standard normal distribution during training. By replacing $p_{\mathcal{X}}(x) = (2\pi)^{-\frac{d}{2}} \exp(-\frac{1}{2}x^T x)$ in Eq. (1), we rewrite the optimization objective in Eq. (2) as follows:

$$\omega^* = \underset{\omega}{\arg\min} \, \mathbb{E}_{q \sim p_Q} \left[ \frac{1}{2}\Phi_\omega(q)^\top \Phi_\omega(q) - \sum_{c=1}^C \log |\det J_{\Phi_c}(k_{c-1})| \right]. \quad (4)$$

Since $d/2\log(2\pi)$ is constant, we remove it in Eq. (4). To learn the ID query feature distribution, we define the maximum likelihood loss function as follows:

$$\mathcal{L}_1 = \mathbb{E}_{q \in Q^{id}} \left[ \frac{1}{2}\Phi_\omega(q)^T \Phi_\omega(q) - \sum_{c=1}^C \log |\det J_{\Phi_c}(k_{c-1})| \right]. \quad (5)$$

## 3.3 OOD Boundary Reasoning

Based on the learned ID query feature distribution, we can search an explicit and compact separating boundary between ID and OOD query. To reduce the computational cost from high-dimensional query features, we consider searching the boundary based on the uncertainty score. Since the log-likelihoods generated by the normalizing flow can be equivalently converted to uncertainty scores, we select the boundary on the log-likelihood distribution.

*3.3.1 Uncertainty Score.* By the normalizing flow, we can estimate the exact log-likelihood $\log p(q)$ for each query feature $q$:

$$\log p(q) = \sum_{c=1}^{C} \log |\det J_{\Phi_c}(k_{c-1})| - \frac{1}{2}\Phi_\omega(q)^T \Phi_\omega(q). \quad (6)$$

With the estimated log-likelihood $\log p(q)$, we utilize the exponential function to convert it to likelihood. Since we aim to maximize log-likelihoods for normal features in Eq. (5), the likelihood can directly measure the uncertainty. Thus, we can obtain the following uncertainty score: $u(q) = \max_{q' \in Q}(\exp(\log p(q'))) - \exp(\log p(q))$, where $u(q)$ is the uncertainty score of query $q$. The log-likelihood can be equivalently converted to the uncertainty score since the exponential function is monotonic. Therefore, the boundary in uncertainty score distribution is equivalent to the separating boundary in log-likelihood distribution.

*3.3.2 Reasoning ID-OOD Separating Boundary.* With log-likelihood distribution, we can search the separating boundary as follows: 1) Firstly, we search the ID log-likelihood distribution by ID log-likelihoods $\mathcal{P}_{id} = \{\log p_i\}_{i=1}^{N_{id}}$ based on the log-likelihood estimation formulation in Eq. (6). 2) Then, we can approximate the log-likelihood distribution of all ID queries by $\mathcal{P}_{id}$. 3) After that, we introduce a position hyper-parameter $\alpha$ to determine the boundary. Specially, we set the $\alpha$-th percentile (*e.g.*, $\alpha = 5$) of sorted ID log-likelihood distribution as the ID boundary $b_{id}$, which also serves as the upper bound of the ID false positive rate is $\alpha\%$. 4) To enhance the robustness of our model, we employ the margin hyper-parameter $\Delta$ and define an OOD boundary as $b_{ood} = b_{id} - \Delta$.

## 3.4 ID-OOD Boundary Refinement

With the explicit ID-OOD boundary, we design an ID-OOD boundary refinement module for discriminative feature learning. Then, we use the boundary $b_{id}$ as the contrastive target. Specially, we push OOD query features whose log-likelihoods are larger than $b_{ood}$ apart from $b_{id}$ at least beyond the margin $\Delta$, while pulling together ID query features whose log-likelihoods are smaller than $b_{id}$. Thus, we introduce the following triplet loss:

$$\mathcal{L}_2 = \sum_{i=1}^{N_{id}} |\min((\log p_i - b_{id}), 0)| + \sum_{j=1}^{N_{ood}} |\max((\log p_j - b_{id} + \Delta), 0)|. \quad (7)$$

Since any log-likelihood $\log p_i$ fallen into the margin region $(b_{ood}, b_{id})$ will increase the value of $\mathcal{L}_2$, we can encourage the sparse log-likelihood distribution in the margin region $(b_{ood}, b_{id})$. The log-likelihoods can range from a large region $(-\infty, 0]$, which makes it difficult to select the satisfactory hyper-parameter $\Delta$. To normalize the log-likelihoods to the small range (*e.g.*, $[-1, 0]$), we introduce a large enough normalizer $h_{id}$ (*e.g.*, $h_{id} = 100$). Since these log-likelihoods can be easily divided into OOD queries, the extremely small log-likelihoods (less than $-1$) can be excluded outside the loss in Eq. (7). Thus, minimizing the loss in Eq. (7) will encourage all log-likelihoods $\mathcal{P}$ to distribute in the regions $[-1, b_{ood}]$ or $[b_{id}, 0]$.

Given a query, if its log-likelihood falls into $[b_{id}, 0]$, we treat it as ID. Otherwise, it is OOD.

## 3.5 Cross-modal Interaction and Training

*3.5.1 Video-query Matching.* The frame-level video and word-level query representations are denoted as $V \in \mathbb{R}^{N_v \times d}$ and $W \in \mathbb{R}^{N_w \times d}$, respectively. Specially, we first extract the global representation by an attentive pooling: $v' = \sum_{i=1}^{N_v} f_i^v v'_i$, $f^v = \xi_{softmax}(VM_v)$, $q' = \sum_{j=1}^{N_w} f_j^q q'_i$, $f^q = \xi_{softmax}(WM_q)$, where $M_v \in \mathbb{R}^{d \times 1}$ and $M_q \in \mathbb{R}^{d \times 1}$ are learnable matrices; $v'$ and $q'$ are global video and query representations, respectively. To evaluate the video-query matching score, we introduce the following cosine similarity:

$$Sim(v', q') = \frac{v'^\top \cdot q'}{\|v'\|_2 \|q'\|_2}, \quad (8)$$

where $\|\cdot\|_2$ denotes the L2-norm of a vector.

In a training batch with $N_b$ video-query pairs $\{V_i, Q_i\}_{i=1}^{N_b}$, we adopt the multi-modal features $\{v'_i, q'_i\}_{i=1}^{N_b}$ for cross-modal semantic alignment. For positive and negative video-query pairs, we utilize the video-query matching score $Sim(v', q')$ for the following alignment loss:

$$\mathcal{L}_3 = -\frac{1}{|N_b|} \sum_{i=1}^{N_b} \log \frac{\exp(Sim(v'_i, q'_i)/\eta)}{\sum_{i \neq j} \exp(Sim(v'_i, q'_j)/\eta)}, \quad (9)$$

where $\eta$ is a temperature parameter.

*3.5.2 Frame-query Matching.* To further better understand the given video $V$ and ID query $Q^{id}$, we design a **F**rame-**Q**uery **M**atching (FQM) module to estimate the matching videos and queries based on the likelihood of each frame about queries.

Specially, the FQM module includes two linear layers followed by two functions (sigmoid and tanh), which is defined as: $FQM(f_i^v, q') = \xi_{sigmoid}(\xi_{tanh}(f_i^v M_1)q' M_2)$, where $\xi_{sigmoid}(\cdot)$ and $\xi_{tanh}(\cdot)$ denote sigmoid and tanh functions, respectively; $f_i^v$ denotes the $i$-th frame feature in video $V$; $M_1 \in \mathbb{R}^{d \times 1}$ and $M_2 \in \mathbb{R}^{d \times 1}$ are learnable matrices. We denote the likelihood of each frame about $q'$ as $P(f_i^v \mid q')$. Also, we can obtain the probability: $P(f_i^v \mid q') = FQM(f_i^v, q')$.

Then, we calculate the frame-aware attention scores $a$ by the softmax function as: $a = \xi_{softmax}(p_1, p_2, ..., p_{N_v})$, where $\xi_{softmax}(\cdot)$ denotes the softmax function. Based on $a$, we can enhance the frame-level feature $f_i^v$: $\hat{f}_i^v = a \odot f_i^v$, where $\odot$ denotes an element-wise product and $\hat{f}_i^v$ is the enhanced frame features. After obtaining $\hat{f}_i^v$, we further integrate it with multi-level query features as: $f_i^{fuse} = M_3 \hat{f}_i^v + M_4 \sum_{j=1}^{N_w} w_j + M_5 q'$, where $f_i^{fuse}$ is the fused feature, and $M_3$, $M_4$ and $M_5$ are learnable weight matrices.

*3.5.3 Positive-unlabeled Learning.* Since most of the video content is background, it is significant to predict the foreground frames that are relevant to the ID query. We can treat query-relevant frames as positive data, whereas query-irrelevant frames are unlabeled data. We can transfer the VMR problem to a semi-supervised learning problem, positive-unlabeled learning (PUL) [1]. Thus, we design a simple yet effective PUL module to predict the target moment.

Specifically, we follow previous VMR works [82] to retrieve the target moment with pre-defined moment proposals based on $f_i^{fuse}$, where we sample $t$ proposals for each frame. Then, we utilize Eq. (8) to obtain the similarity score $s_i \in [0, 1]$ between the ID query

and the $i$-th proposal. In a training batch $\mathcal{S} = \{s_i\}$, we can divide these proposals into two sets: positive set $\overline{\mathcal{P}} = \{s_i | s_i \geq 0.5\}$ and the unlabeled background set $\overline{\mathcal{U}} = \{s_i | s_i < 0.5\}$. Therefore, we ascendingly sort $\overline{\mathcal{U}}$ and select top-$N_s$ samples to form the most likely negative set $\overline{\mathcal{N}} = \{s_i | s_i \in sort(\overline{\mathcal{U}})_{1,...,N_s}\}$. Since $|\overline{\mathcal{U}}| \gg |\overline{\mathcal{P}}|$ in most batches, we set $N_s = |\overline{\mathcal{N}}| := \min(|\overline{\mathcal{P}}|, |\overline{\mathcal{U}}|)$ for smooth training. Finally, we employ the binary cross-entropy loss based on the positive set $\overline{\mathcal{P}}$ and the negative set $\overline{\mathcal{N}}$:

$$\mathcal{L}_{\text{BCE}} = -\frac{1}{|\overline{\mathcal{P}}|} \sum_{s_i \in \overline{\mathcal{P}}} \log s_i - \frac{1}{|\overline{\mathcal{N}}|} \sum_{s_i \in \overline{\mathcal{N}}} \log(1 - s_i). \quad (10)$$

By Eq. (10), we can push the probably pure background proposals far away from positive proposals. Although the PUL method is simple, the learned similarity scores are enough for distinguishing the foreground and background proposals.

As the boundaries of pre-defined proposals are coarse, we employ a retrieval regression loss for calibrating the retrieval. We calculate the regression loss for every positive sample:

$$\mathcal{L}_{\text{reg}} = \frac{1}{|\overline{\mathcal{P}}|} \sum \mathcal{L}_{smooth}(t_s, t'_s) + \mathcal{L}_{smooth}(t_e, t'_e), \quad (11)$$

where $\mathcal{L}_{smooth}$ is the smooth $L_1$ loss; $t_s, t_e$ are the ground-truth start and end timestamps; $t'_s, t'_e$ are the predicted timestamps.

For convenience, we set $\mathcal{L}_4 = \mathcal{L}_{\text{BCE}} + \mathcal{L}_{\text{reg}}$. Therefore, we can obtain the following overall loss:

$$\mathcal{L} = \mathcal{L}_1 + \lambda_1 \mathcal{L}_2 + \lambda_2 \mathcal{L}_3 + \lambda_3 \mathcal{L}_4, \quad (12)$$

where $\lambda_1, \lambda_2$ and $\lambda_3$ are the balanced weight hyper-parameters.

**Inference.** Given a video and a sentence query, we first feed them into our model to detect if the query is ID or OOD. If the query is ID, we can obtain the fused cross-modal feature $f_i^{fuse}$. Then, we predict the moment boundary $(t'_s, t'_e)$ in Eq. (11). "Top-n (R@n)" moment candidates will be selected with non-maximum suppression [44].

## 4 EXPERIMENTS

### 4.1 Datasets and Evaluation Metrics

**Datasets.** We experiment on four VMR datasets with various characteristics: Charades-STA [12], Activitynet-Captions [25]and TACoS [50]. In this paper, we conduct experiments under two settings: closed-set and open-set, whose split principle is illustrated in Table 1. In the closed-set setting, for each dataset, we utilize its queries and its videos for both training and testing. In the open-set setting, we train each dataset based on its queries and videos, while we conduct testing on its videos and queries from all three datasets.

**ActivityNet Captions.** ActivityNet Captions is introduced by [25], which contains about 20k untrimmed videos and 100k descriptions with diverse open-domain activities. Following the split principle in [81], in the closed setting, we utilize 37,417 video-query pairs for training, and 34,536 pairs for testing, respectively.

**Charades-STA.** It is built on Charades [53] by [12], including 9,848 videos of indoor scenarios. In the closed setting, we use 12,408 and 3,720 video-sentence pairs for training and testing, respectively.

**TACoS.** TACoS is a challenging dataset collected from the MPII Cooking Composite Activities [51], which contains 127 long videos of cooking scenarios. In the closed setting, we follow the standard split used in [12] and obtain 10,146, and 8,672 video-sentence pairs as training and testing dataset, respectively.

**Table 1: Closed-set datasets and open-set datasets during training and testing, where "ANC" denotes "ActivityNet Captions", "CS" denotes "Charades-STA", and "All" denotes all three datasets.**

| Dataset | Closed-set train | | Closed-set test | | Open-set train | | Open-set test | |
|---|---|---|---|---|---|---|---|---|
| | Video | Query | Video | Query | Video | Query | Video | Query |
| ANC | ANC | ANC | ANC | ANC | ANC | ANC | ANC | **All** |
| CS | CS | CS | CS | CS | CS | CS | CS | **All** |
| TACoS | TACoS | TACoS | TACoS | TACoS | TACoS | TACoS | TACoS | **All** |

**Evaluation metrics.** There are two challenges in our method: *ID query for moment retrieval* and *OOD query detection*. Thus, we use two types of metrics for comprehensive performance evaluation:

1) *ID query for moment retrieval.* Following [12, 79], we adopt "R@n, IoU=m" as the evaluation metric, which denotes the percentage of language queries having at least one result whose Intersection over Union (IoU) with ground truth is larger than m in top-n retrieved moment. In our experiments, we use $n \in \{1, 5\}$ for all datasets, $m \in \{0.5, 0.7\}$ for Charades-STA, $m \in \{0.3, 0.5\}$ for ActivityNet Captions and TACoS.

2) *OOD query detection.* To evaluate open-set performance, we introduce two popular metrics to evaluate the performance of detecting ID and OOD queries for a given video: the Area Under the Receiver Operating Characteristic (**AUROC**) curve and the Area Under the Precision-Recall (**AUPR**) [37].

For all the metrics, the larger value denotes better performance.

**Implementation details.** For the video encoder, we apply C3D [58] to encode the videos on ActivityNet Caption and I3D [2] on Charades-STA and TACoS. Since some videos are overlong, we set the length of frame sequences $M$ to 64, 64 and 200 for Charades-STA, ActivityNet Captions and TACoS, respectively. For the query encoder, we utilize GloVe 840B 300d [48] to embed each word as word features. We sample 800 moment proposals for TACoS and 384 for Charades-STA and ActivityNet Captions similar with [82]. We train our model for 200 epochs with a batch size of 128 and an early stopping strategy. Parameter optimization is performed by Adam [22] optimizer with a learning rate of 0.0008. We conduct our experiments on a single Nvidia TITAN XP GPU.

### 4.2 Comparison with State-of-the-Arts

*4.2.1 Compared Methods.* We conduct performance comparison on all three datasets under both closed-set and open-set settings. To evaluate efficiency, we only choose the open-source compared methods that are grouped into two categories: (i) **Fully-supervised (FS)** setting [12–14, 30, 31, 34, 64, 72, 75, 80–82]; (ii) **Weakly-supervised (WS)** setting [4, 62, 63, 65, 83, 85]. For convenience, we denote our proposed "**open-set setting**" as "**OS**". Following [42, 78], we directly cite the results of compared methods from corresponding works. Note that no weakly-supervised method reports its results on TACoS. The best results are **bold**.

*4.2.2 Closed-set Evaluation.* The quantitative comparison results of our model and compared methods on ActivityNet Captions, Charades-STA, and TACoS Captions are reported in Table 2, 3 and 4, respectively.

Based on these experimental results, we list several notable observations as follows: 1) Compared with other state-of-the-art methods, our proposed model achieves the highest performance over all metrics on three datasets, which demonstrates the superiority

**Table 2: Effectiveness comparison for closed-set VMR on ActivityNet Captions dataset under official train/test splits.**

| Method | Type | R@1, IoU=0.3 | R@1, IoU=0.5 | R@5, IoU=0.3 | R@5, IoU=0.5 |
|---|---|---|---|---|---|
| CTRL [12] | FS | - | 29.01 | - | 59.17 |
| 2D-TAN [81] | FS | 59.45 | 44.51 | 85.53 | 77.13 |
| DRN [75] | FS | - | 45.45 | - | 77.97 |
| RaNet [13] | FS | - | 45.59 | - | 75.93 |
| MIGCN [80] | FS | - | 48.02 | - | 78.02 |
| MMN [64] | FS | 65.05 | 48.59 | 87.25 | 79.50 |
| G2L [30] | FS | - | 51.68 | - | 81.32 |
| ICVC [4] | WS | 46.62 | 29.52 | 80.92 | 66.61 |
| LCNet [65] | WS | 48.49 | 26.33 | 82.51 | 62.66 |
| VCA [63] | WS | 50.45 | 31.00 | 71.79 | 53.83 |
| WSTAN [62] | WS | 52.45 | 30.01 | 79.38 | 63.42 |
| CNM [85] | WS | 55.68 | 33.33 | - | - |
| **Ours** | **OS** | **69.85** | **56.47** | **94.38** | **86.54** |

**Table 3: Performance comparison for closed-set VMR on Charades-STA dataset under official train/test splits.**

| Method | Type | R@1, IoU=0.5 | R@1, IoU=0.7 | R@5, IoU=0.5 | R@5, IoU=0.7 |
|---|---|---|---|---|---|
| CTRL [12] | FS | 23.62 | 8.89 | 58.92 | 29.52 |
| MMN [64] | FS | 47.31 | 27.28 | 83.74 | 58.41 |
| 2D-TAN [81] | FS | 39.81 | 23.25 | 79.33 | 52.15 |
| RaNet [13] | FS | 43.87 | 26.83 | 86.67 | 54.22 |
| DRN [75] | FS | 45.40 | 26.40 | 88.01 | 55.38 |
| G2L [30] | FS | 47.91 | 28.42 | 84.80 | 59.33 |
| MomentDiff [31] | FS | 53.79 | 30.18 | - | - |
| WSTAN [62] | WS | 29.35 | 12.28 | 76.13 | 41.53 |
| ICVC [4] | WS | 31.02 | 16.53 | 77.53 | 41.91 |
| CNM [85] | WS | 35.15 | 14.95 | - | - |
| VCA [63] | WS | 38.13 | 19.57 | 78.75 | 37.75 |
| LCNet [65] | WS | 39.19 | 18.17 | 80.56 | 45.24 |
| **Ours** | **OS** | **58.38** | **33.53** | **93.62** | **62.35** |

**Table 4: Performance comparison for closed-set VMR on TACoS dataset under official train/test splits.**

| Method | Type | R@1, IoU=0.3 | R@1, IoU=0.5 | R@5, IoU=0.3 | R@5, IoU=0.5 |
|---|---|---|---|---|---|
| CTRL [12] | FS | 18.32 | 13.30 | 36.69 | 25.42 |
| ACRN [34] | FS | 19.52 | 14.62 | 34.97 | 24.88 |
| CMIN [82] | FS | 24.64 | 18.05 | 38.46 | 27.02 |
| SCDM [72] | FS | 26.11 | 21.17 | 40.16 | 32.18 |
| DRN [75] | FS | - | 23.17 | - | 33.36 |
| 2D-TAN [81] | FS | 37.29 | 25.32 | 57.81 | 45.04 |
| MMN [64] | FS | 39.24 | 26.17 | 62.03 | 47.39 |
| FVMR [14] | FS | 41.48 | 29.12 | 64.53 | 50.00 |
| G2L [30] | FS | 42.74 | 30.95 | 65.83 | 49.86 |
| RaNet [13] | FS | 43.34 | 33.54 | 67.33 | 55.09 |
| MomentDiff [31] | FS | 44.78 | 33.68 | - | - |
| **Ours** | **OS** | **55.44** | **42.72** | **73.48** | **64.48** |

**Table 5: Effectiveness comparison for open-set VMR on ActivityNet Captions under official train/test splits, where Ours(a), Ours(b) and Ours(c) are our three ablation models, Ours(full) is our full model.**

| Method | Type | R@1, IoU=0.3 | R@1, IoU=0.5 | AUROC | AUPR |
|---|---|---|---|---|---|
| CTRL [12] | FS | 22.16 | 13.42 | - | - |
| 2D-TAN [81] | FS | 30.88 | 22.95 | - | - |
| DRN [75] | FS | 31.75 | 26.34 | - | - |
| RaNet [13] | FS | 30.96 | 28.43 | - | - |
| MIGCN [80] | FS | 32.40 | 29.57 | - | - |
| MomentDiff [31] | FS | 29.59 | 30.17 | - | - |
| MMN [64] | FS | 35.17 | 29.92 | - | - |
| ICVC [4] | WS | 18.43 | 10.76 | - | - |
| LCNet [65] | WS | 20.48 | 11.69 | - | - |
| VCA [63] | WS | 21.50 | 10.36 | - | - |
| WSTAN [62] | WS | 24.76 | 12.38 | - | - |
| CNM [85] | WS | 28.44 | 15.74 | - | - |
| **Ours(a)** | **OS** | 42.15 | 30.03 | 33.20 | 38.87 |
| **Ours(b)** | **OS** | 45.02 | 31.72 | 32.48 | 37.39 |
| **Ours(c)** | **OS** | 48.68 | 36.84 | 38.16 | 41.80 |
| **Ours(full)** | **OS** | **50.96** | **38.75** | **39.24** | **43.68** |

**Table 6: Performance comparison for open-set VMR on Charades-STA under official train/test splits, where Ours(a), Ours(b) and Ours(c) are our three ablation models, Ours(full) is our full model.**

| Method | Type | R@1, IoU=0.5 | R@1, IoU=0.7 | AUROC | AUPR |
|---|---|---|---|---|---|
| CTRL [12] | FS | 9.16 | 2.35 | - | - |
| MMN [64] | FS | 21.85 | 10.43 | - | - |
| 2D-TAN [81] | FS | 16.46 | 8.13 | - | - |
| MomentDiff [31] | FS | 12.68 | 7.26 | - | - |
| RaNet [13] | FS | 18.50 | 9.77 | - | - |
| DRN [75] | FS | 20.13 | 8.10 | - | - |
| WSTAN [62] | WS | 8.15 | 3.92 | - | - |
| ICVC [4] | WS | 10.40 | 2.97 | - | - |
| CNM [85] | WS | 11.43 | 6.28 | - | - |
| VCA [63] | WS | 14.72 | 5.71 | - | - |
| LCNet [65] | WS | 13.52 | 7.39 | - | - |
| **Ours(a)** | **OS** | 28.40 | 16.77 | 32.02 | 35.63 |
| **Ours(b)** | **OS** | 30.27 | 15.29 | 33.82 | 34.95 |
| **Ours(c)** | **OS** | 33.18 | 19.05 | 36.75 | 39.86 |
| **Ours(full)** | **OS** | **34.62** | **19.58** | **38.76** | **40.29** |

of our proposed model. 2) On the ActivityNet Captions dataset, our model outperforms all the compared methods over all metrics. Particularly, our model beats the best compared fully-supervised method G2L by 5.22% in terms of "R@5, IoU=0.5". As for the best compared weakly-supervised method CNM, our model outperforms it by 14.17% over "R@1, IoU=0.3". It shows that our model has excellent generalization ability in more complex and diverse real-world scenarios by frame-query matching and video-query matching. 3) On the Charades-STA dataset, based on the reported settings of

previous methods, our proposed method reaches the new state-of-the-art on all the metrics. In terms of "R@1, IoU=0.3", our model outperforms the best compared fully-supervised method Moment-Diff by 4.59%, and the best compared weakly-supervised method LCNet by 19.19%. 4) On the TACoS dataset, our model surpasses other state-of-the-art methods by a large margin. Compared with best compared method, our model improves the performance by 10.66% and 9.04% in terms of "R@1, IoU=0.3" and "R@1, IoU=0.5", which may stem from the inherent characteristics of this dataset.

*4.2.3 Open-set Evaluation.* Similarly, we conduct open-set experiments on all three datasets. Table 5, 6 and 7 illustrate the open-set VMR performance on ActivityNet Captions, Charades-STA, and TACoS, respectively. Obviously, our model achieves the best performance on three datasets over all the metrics.

**Open-set results on ActivityNet Captions:** We report its open-set comparison in Table 5. Particularly, our model beats the best compared fully-supervised method MMN by 15.79% in terms of "R@1, IoU=0.5". As for the best compared weakly-supervised method

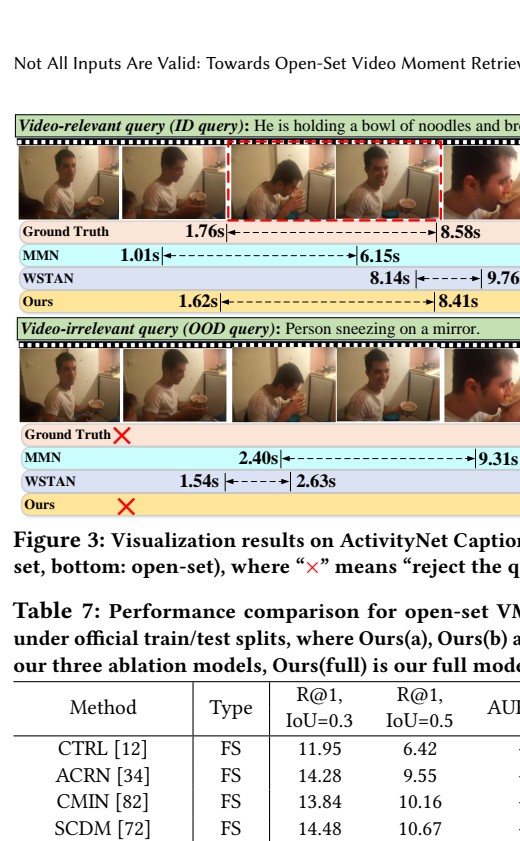

**Figure 3: Visualization results on ActivityNet Captions (top: closed-set, bottom: open-set), where "✗" means "reject the query".**

**Table 7: Performance comparison for open-set VMR on TACoS under official train/test splits, where Ours(a), Ours(b) and Ours(c) are our three ablation models, Ours(full) is our full model.**

| Method | Type | R@1, IoU=0.3 | R@1, IoU=0.5 | AUROC | AUPR |
|---|---|---|---|---|---|
| CTRL [12] | FS | 11.95 | 6.42 | - | - |
| ACRN [34] | FS | 14.28 | 9.55 | - | - |
| CMIN [82] | FS | 13.84 | 10.16 | - | - |
| SCDM [72] | FS | 14.48 | 10.67 | - | - |
| DRN [75] | FS | 15.73 | 11.05 | - | - |
| 2D-TAN [81] | FS | 14.80 | 11.13 | - | - |
| MMN [64] | FS | 17.64 | 11.52 | - | - |
| FVMR [14] | FS | 14.83 | 10.10 | - | - |
| MomentDiff [31] | FS | 16.28 | 12.34 | - | - |
| RaNet [13] | FS | 13.72 | 9.45 | - | - |
| MIGCN [80] | FS | 14.13 | 11.39 | - | - |
| **Ours(a)** | OS | 33.81 | 19.76 | 38.63 | 34.82 |
| **Ours(b)** | OS | 35.48 | 20.87 | 37.31 | 35.29 |
| **Ours(c)** | OS | 36.54 | 22.47 | 40.82 | 35.96 |
| **Ours(full)** | **OS** | **38.27** | **23.60** | **41.95** | **37.80** |

CNM, our model outperforms it by 23.01% over "R@1, IoU=0.3". The main reason is that compared methods cannot distinguish ID and OOD queries, and directly utilize OOD queries for VMR, which significantly reduces their performance under the realistic open-set setting. For the challenging open-set VMR task, we can correctly recognize OOD queries and precisely retrieve the target moment by ID queries, which illustrates the effectiveness of our method.

**Open-set results on Charades-STA:** In Table 6, we further evaluate our open-set performance on the Charades-STA dataset. Our method achieves the best results in all the cases. For example, our model outperforms the best compared fully-supervised method DRN by 14.49% and 11.48% in terms of "R@1, IoU=0.3" and "R@1, IoU=0.5", respectively. Besides, our model surpasses the best weakly-supervised LCNet by 21.10% and 12.19% over "R@1, IoU=0.3" and "R@1, IoU=0.5", respectively. This is because the Charades-STA dataset is under the indoor scenarios, which are irrelevant to outdoor queries in other two datasets. Previous methods directly retrieve outdoor moments based on outdoor queries, leading to unreasonable retrieval results for the open-set VMR task.

**Open-set results on TACoS:** As shown in Table 7, we also evaluate the open-set performance of our model on the TACoS dataset.

**Table 8: Efficiency comparison for closed-set VMR on TACoS.**

| Method | Run-Time | Model Size | R@1, IoU=0.5 |
|---|---|---|---|
| ACRN [34] | 5.96s | 128M | 14.62 |
| CTRL [12] | 3.58s | **22M** | 13.30 |
| TGN [3] | 0.89s | 166M | 18.90 |
| 2D-TAN [81] | 0.71s | 232M | 25.32 |
| DRN [75] | 0.22s | 214M | 23.17 |
| MomentDiff [31] | 1.85s | 248M | 33.68 |
| **Ours** | **0.08s** | 92M | **39.72** |

Particularly, our model outperforms the best compared method MomentDiff by 24.14% and 12.21% over "R@1, IoU=0.3" and "R@1, IoU=0.5", respectively. The main reason for the significant performance improvement is that all the videos on TACoS are cooking-related, which only corresponds to cooking-related queries. Given some cooking-irrelevant OOD queries from other two datasets, previous methods mistakenly utilize these OOD queries for wrong moment retrieval, which severely limits their performance.

*4.2.4 Efficiency Comparison.* In Table 8, we evaluate the efficiency of our proposed model, by fairly comparing its running time and model size in the inference phase with existing open-source methods on TACoS. Obviously, we achieve much faster processing speeds with relatively fewer learnable parameters. This attributes to: 1) Proposal-based methods (ACRN, CTRL, TGN, 2D-TAN, DRN) suffer from the time-consuming proposal-matching process. Unlike them, our retrieval module utilizes the positive-unlabeled learning module, which is much more efficient. 2) Different from them, our model only learns an effective and efficient retrieval backbone without introducing additional parameters during inference.

*4.2.5 Visualization.* Fig. 3 depicts the retrieval visualizations on ActivityNet Captions under both closed-set and open-set settings. In the closed-set setting, our model achieves better retrieval performance than previous state-of-the-art methods (MMN and WSTAN). This is because our model can fully conduct cross-modal interaction by both video-query matching and frame-query matching. For the challenging open-set setting, our model can reason the right ID-OOD boundary for OOD query recognition, thus rejecting video-irrelevant queries.

## 4.3 Ablation Study and Analysis

**Main ablation studies.** To demonstrate the effectiveness of each component in our model, we conduct ablation studies regarding the components (*i.e.*, ID knowledge acquisition module, uncertainty-aware OOD boundary reasoning module and ID-OOD boundary refinement module). In particular, we remove each key individual module to investigate its contribution. For convenience, we design three ablation models: 1) Ours(a). We remove the ID knowledge acquisition module while keeping the other two modules. 2) Ours(b). We remove the uncertainty-aware OOD boundary reasoning module while keeping the other two modules. 3) Ours(c). We remove the ID-OOD boundary refinement module while keeping the other two modules. Besides, we use our full model as the baseline: Ours(full). Since we focus on the open-set setting, we conduct corresponding experiments in Table 5, 6 and 7. Based on these tables, we can observe that all three modules contribute a lot to the final performances on all three datasets, demonstrating their effectiveness in the open-set VMR task. As the core module for detecting OOD

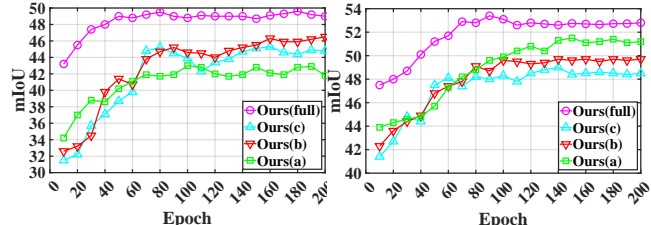

**Figure 4: Training performance of each ablation module for open-set VMR on ActivityNet Captions (left) and Charades-STA (right).**

**Table 9: Effect of ID Knowledge Acquisition (IKA), Uncertainty-aware OOD Boundary Reasoning (UOBR) and ID-OOD Boundary Refinement (IBR) for open-set VMR on Charades-STA, where "DD" means "Dirichlet Distribution", "MGD" means "Multi-variate Gaussian Distribution" and "MD" means "Multinomial Distribution".**

| Module | Changes | R@1, IoU=0.5 | R@1, IoU=0.7 | AUROC | AUPR |
|---|---|---|---|---|---|
| IKA | C=5 | 33.87 | 19.21 | 38.08 | 39.22 |
| | **C=6** | **34.62** | **19.58** | 38.76 | **40.29** |
| | C=7 | 34.18 | 19.04 | **38.96** | 40.23 |
| | DD | 33.14 | 16.38 | 35.71 | 38.55 |
| | **MGD** | **34.62** | **19.58** | **38.76** | **40.29** |
| | MD | 31.42 | 15.80 | 34.33 | 39.12 |
| UOBR | $\alpha = 4$ | 34.24 | 18.52 | 38.01 | 40.10 |
| | $\alpha = 5$ | **34.62** | **19.58** | **38.76** | **40.29** |
| | $\alpha = 6$ | 33.73 | 19.41 | 37.95 | 40.20 |
| IBR | $\Delta = 0.03$ | 34.23 | 18.02 | 37.15 | 39.86 |
| | $\Delta = 0.04$ | **34.62** | **19.58** | **38.76** | **40.29** |
| | $\Delta = 0.05$ | 34.34 | 18.60 | 38.12 | 39.39 |

queries, the ID knowledge acquisition module brings the greatest improvement, illustrating that it provides ID distribution information for ID boundary reasoning. Also, the uncertainty-aware OOD boundary reasoning module achieves significant performance improvement, which illustrates the effectiveness of our uncertainty score and OOD boundary reasoning. Besides, the ID-OOD boundary refinement module improves performance in all metrics.

**Training process of different ablation models.** Following [33], we analyze the training process and retrieval performance of different ablation models in Fig. 4. We can obtain the following representative observations: (i) During training, Our(full) outperforms other ablation models, which further demonstrates the effectiveness of each module. (ii) Our(full) converges faster than ablation models, which shows that our full model is more efficient on time-consuming. For instance, Our(full) converges within 120 epochs on both ActivityNet Captions and Charades-STA, while Our(c) converges after 160 epochs. Thus, our full model can process these challenging datasets more efficiently.

**Effect of ID knowledge acquisition.** In the ID knowledge acquisition (IKA) module, we design a multi-layer coupling block and utilize the multi-variate Gaussian distribution (MGD). We implement different variants of the IKA module in Table 9. Obviously, we can achieve the best performance when we utilize the five-layer coupling block and MGD to learn ID distribution. It demonstrates that the five-layer coupling block can sufficiently construct the normalizing flow and learn ID query feature distribution based on multi-variate Gaussian distribution assumption.

**Analysis on uncertainty-aware OOD boundary reasoning.** Similarly, we conduct ablation on the uncertainty-aware OOD

**Table 10: Ablations on cross-modal interaction for open-set VMR on ActivityNet Captions, where "VQM" means "video-query matching", "FQM" means "frame-query matching", "PUL" means "positive-unlabeled learning".**

| Ablation | R@1, IoU=0.5 | R@1, IoU=0.7 | AUROC | AUPR |
|---|---|---|---|---|
| w/o VQM | 47.93 | 35.40 | 35.92 | 41.25 |
| w/o FQM | 48.50 | 37.02 | 37.06 | 42.54 |
| w/o PUL | 48.17 | 36.64 | 36.56 | 41.86 |
| **Ours(full)** | **50.96** | **38.75** | **39.24** | **43.68** |

**Table 11: Parameter analysis on ActivityNet Captions.**

| Parameter | Changes | R@1, IoU=0.5 | R@1, IoU=0.7 | AUROC | AUPR |
|---|---|---|---|---|---|
| $\lambda_1$ | 0.5 | 50.28 | 38.12 | 38.06 | 42.39 |
| | **0.6** | **50.96** | **38.75** | **39.24** | **43.68** |
| | 0.7 | 49.81 | 38.10 | 38.94 | 43.51 |
| $\lambda_2$ | 0.2 | 50.26 | **38.89** | 38.72 | 40.13 |
| | **0.3** | **50.96** | 38.75 | **39.24** | **43.68** |
| | 0.4 | 50.11 | 38.24 | 38.50 | 42.18 |
| $\lambda_3$ | 0.6 | 50.16 | 37.87 | 38.25 | 42.75 |
| | **0.7** | 50.96 | 38.75 | 39.24 | **43.68** |
| | 0.8 | **51.29** | **38.86** | 38.95 | 42.94 |
| $\eta$ | 0.1 | 50.36 | 38.28 | 38.77 | **43.92** |
| | **0.2** | **50.96** | **38.75** | **39.24** | 43.68 |
| | 0.3 | 50.29 | 38.15 | 38.76 | 43.10 |

boundary reasoning (UOBR) module. As shown in Table 9, we change the value of $\alpha$ to search the best results. Thus, we set $\alpha = 5$ in all the experiments.

**Influence of ID-OOD boundary refinement.** Also, we conduct ablation study on the ID-OOD Boundary Refinement module. Table 9 illustrates the ablation results on the Charades-STA dataset. Obviously, when $\Delta = 0.04$, we can obtain the best performance.

**Analysis on cross-modal interaction.** As shown in Table 10, we further evaluate the importance of three components in the cross-modal interaction module: video-query matching, frame-query matching and positive-unlabeled learning. We find that the video-query matching component achieves the largest improvement, which is because wrong video-query matching will directly lead to retrieval failure. Moreover, positive-unlabeled learning brings significant improvement since it can recognize the proper moment proposals by the binary cross-entropy loss. In addition, frame-query matching provides fine-grained vision-language alignment for more precise retrieval. Thus, all three components in our cross-modal interaction module are significant.

**Parameter analysis.** As shown in Table 11, we conduct experiments on the ActivityNet Captions dataset under the open-set setting, and present the ablation study on the hyper-parameters $(\lambda_1, \lambda_2, \lambda_3, \eta)$. We can find that, their performance only varies in a small range, indicating that our model is insensitive to these parameters. Finally, we choose $\lambda_1 = 0.6, \lambda_2 = 0.3, \lambda_3 = 0.7, \eta = 0.2$.

## 5 CONCLUSION

In this paper, we pose a brand-new and challenging task: open-set video moment retrieval (OS-VMR). Given a video and a query, the OS-VMR task aims to conduct retrieval if the query is video-relevant, otherwise reject the query. To address it, we propose a novel method for this special task. Experiments on three challenging benchmarks demonstrate the effectiveness of our method.

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
