# OpenReview forum: "Not All Inputs Are Valid: Towards Open-Set Video Moment Retrieval using Language"
_acmmm.org/ACMMM/2024/Conference — MM2024 Oral_

### Official Review · Reviewer_tnpK · 2024-04-29

**Rating:** 2
**Confidence:** 4

**Summary:**

This paper introduces open-set video moment retrieval, which retrieves specific moments based on query sentences and can reject OOD queries. The proposed model uses normalizing flow technology to distinguish between ID and OOD queries.  Moreover, it refines the ID-OOD boundary and incorporates cross-modal interactions for video-query and frame-query matching. Additionally, a positive-unlabeled learning module is introduced for moment retrieval. Experimental results on three datasets demonstrate its effectiveness.

**Strengths:**

1. This paper proposes the open-set Video Moment Retrieval (OS-VMR) task, which is interesting and valuable in open-set settings.

2. This paper presents an OpenVMR framework to distinguish ID and OOD queries and utilizes ID queries for moment retrieval.

3. Extensive experiments are conducted on three popular datasets.

**Limitations:**

1. How to construct OOD queries?

2. The initial focus of this paper appears to address an OOD detection challenge, suggesting numerous methods could be employed. Why does the paper opt for normalizing flow? Are there alternative strategies considered?

3. The ID queries and video matching rely on commonly employed modules, lacking novel technical insights. Similarly, algorithms for the OOD and ID separation appear conventional. Overall, the paper's contribution appears limited.

4. While the consideration of OOD queries is pertinent, why restrict the discussion solely to OOD queries rather than also addressing OOD videos?

**Suitability:**

3

---

### Official Review · Reviewer_ygKs · 2024-05-24

**Rating:** 3
**Confidence:** 3

**Summary:**

This paper addresses the limitations of existing Video Moment Retrieval (VMR) methods that operate under a closed-set assumption, assuming all input queries are relevant to the video content. The authors introduce a novel task, Open-Set Video Moment Retrieval (OS-VMR), which aims to not only retrieve video moments for in-distribution (ID) queries but also reject out-of-distribution (OOD) queries. They propose a model named OpenVMR, which employs normalizing flow technology to distinguish between ID and OOD queries and subsequently conducts moment retrieval for ID queries. The model involves several key components: learning the ID query distribution, introducing an uncertainty score to separate ID and OOD queries, refining the ID-OOD boundary using a triplet loss, and performing video-query and frame-query matching for accurate moment retrieval. Extensive experiments on three datasets (ActivityNet Captions, Charades-STA, and TACoS) demonstrate the effectiveness of OpenVMR, showing significant improvements over state-of-the-art methods in both closed-set and open-set settings.

**Strengths:**

This paper introduces the novel task of Open-Set Video Moment Retrieval (OS-VMR), addressing an important gap in the current VMR research. By moving beyond the closed-set assumption, the authors tackle a significant real-world problem where irrelevant queries may be presented. The use of normalizing flow technology to differentiate between in-distribution (ID) and out-of-distribution (OOD) queries is innovative and well-executed. Additionally, the comprehensive evaluation across multiple challenging datasets, including ActivityNet Captions, Charades-STA, and TACoS, provides robust empirical evidence supporting the proposed method's effectiveness. The model's performance improvements over state-of-the-art methods are particularly noteworthy.

**Limitations:**

However, the paper is dense and challenging to follow in some sections, particularly in the methodological explanations. Improving clarity and structuring the content more logically could enhance readability. While the paper compares its results with state-of-the-art methods, it would benefit from a more detailed analysis of why the proposed method outperforms these baselines. Providing more insights into the failure modes of existing methods when dealing with OOD queries could strengthen the argument.

The model's performance is impressive on the selected datasets, but the paper lacks a discussion on the generalizability of the approach to other types of videos or queries outside the tested domains. Although the paper includes ablation studies, they could be more detailed. Specifically, analyzing the contribution of each component in varying contexts and datasets would provide a deeper understanding of the model’s robustness.

The paper lacks supplementary materials that could provide additional insights and experimental results. Including detailed experiment logs, hyperparameter configurations, and extended results in the supplementary materials would enhance the transparency and reproducibility of the study. This is a common issue among the state-of-the-art methods cited, which often lack comprehensive supplementary materials, making it difficult to fully validate and reproduce the reported results. Addressing this gap would not only strengthen the current paper but also set a higher standard for future research in this area.

**Suitability:**

3

---

### Official Review · Reviewer_tEHt · 2024-05-26

**Rating:** 5
**Confidence:** 3

**Summary:**

This paper poses a novel and realistic multi-modal task, open-set video moment retrieval, which aims to not only retrieve the precise moments based on ID query, but also to reject OOD queries. Unlike previous methods based on a closed-set assumption that all the given queries as ID, the proposed model OpenVMR can first distinguish ID and OOD queries based on the normalizing flow technology, and then conduct moment retrieval based on ID queries. Extensive experiments on open-set and closed-set settings show the effectiveness of OpenVMR.

**Strengths:**

1. The open-set setting is interesting and novel. The motivation for the open-set setting is clear, and the proposed method is easy to understand. The technical novelty is enough, and the proposed model is sound.

2. This paper is well-written and easy to follow, and the structure is carefully organized. Figures and tables are well-illustrated. The experimental results are impressive.

3. Extensive experiments under both closed-set and open-set settings are conducted. For each core module, the corresponding ablation study is well-designed to fully analyze its effectiveness. Experimental results show that the proposed method can beat other state-of-the-art approaches.

**Limitations:**

1. The authors employ GloVe to extract the textual feature. I am wondering if transformer-based contextualized embedding models, such as BERT, can perform better.

2. In Section 3.2.2, the authors assume that p_X(x) obeys the Multi-variate Gaussian Distribution. Why? Have the authors tried any other multi-variate distributions? e.g., Multivariate Dirichlet Distribution

**Suitability:**

3

---

### Official Review · Reviewer_xgkW · 2024-05-27

**Rating:** 6
**Confidence:** 3

**Summary:**

Video Moment Retrieval (VMR) is a challenging multimedia task that aims to localize a specific moment based on an untrimmed video and a sentence query. Existing VMR methods refer to a close-set assumption that all the queries are video-relevant (ID). However, the authors notice that real-world users might provide a video-irrelevant query (OOD query) in realistic applications, which will lead to unreasonable results. Based on this, the authors wisely pose a novel setting, open-set VMR, which first distinguishes ID and OOD queries, and then conducts moment retrieval based on ID queries. Extensive experiments demonstrate the effectiveness of the proposed method.

**Strengths:**

+The paper is easy to understand, and the methodology is technologically sound. The normalizing flow can learn the query distribution, and the multi-variate Gaussian distribution is very consistent with real-world applications. Authors equivalently convert the log-likelihood to the uncertainty score u(q) for simplifying the computation, which is very ingenious to reason the ID-OOD boundary.

+The experiments are sufficient. The performance improvement is very impressive, which illustrates the effectiveness of the proposed method. Both close-set and open-set settings are considered in this paper. Besides, the ablation study is profound.

+The open-set VMR task is very interesting, novel and consistent with reality. Previous methods mechanically utilize the given query for VMR.

**Limitations:**

-Authors should try other pretrained video or query encoders in Multimodal LLM for better performance.

-Could the proposed method serve as a plug-and-play module for existing VMR works? If the proposed method can help previous works deal with the open-set problem, it will be very exciting.

-Some VMR related work should be included for comparison and analysis, such as “Video Moment Retrieval With Cross-Modal Neural Architecture Search, TIP 2022 ”, “Deconfounded Video Moment Retrieval with Causal Intervention, SIGIR 2021”

**Suitability:**

3

---

### Official Review · Reviewer_fcNQ · 2024-06-04

**Rating:** 4
**Confidence:** 4

**Summary:**

● This paper focuses on Video Moment Retrieval (VMR), which involves retrieving relevant video segments or moments corresponding to natural language queries.
● Specifically, it addresses the challenges of identifying in-domain (ID) queries and out-of-domain (OOD) queries, as well as refining the boundary between them for more accurate retrieval.
● The paper proposes a comprehensive framework comprising several modules, including ID knowledge acquisition, uncertainty-aware OOD boundary reasoning, and ID-OOD boundary refinement.

**Strengths:**

○ The paper introduces the novel concept of Open-set Video Moment Retrieval (OS-VMR), which extends traditional video moment retrieval by requiring the system to handle out-of-domain (OOD) queries in addition to in-domain (ID) queries. This innovation is particularly significant for applications in open-set environments.
○ The paper provides a clear theoretical explanation of how the OpenVMR framework operates.
○ The paper conducts several experiments that validate the effectiveness of the proposed method, demonstrating its ability.

**Limitations:**

Need for Additional Supporting Material:
1.   On line 89, why can this solution be directly obtained?   The author needs to provide more supporting material to substantiate this claim.   Obviously, most of the video content is irrelevant to the query, and only a very short video segment matches the query.
Misunderstandings in the Experimental Section Tables:
2.   Table 11: In the parameter ablation experiment, what are the default values of the other parameters when one parameter is fixed?   The author needs to clarify.
3.   Table 10: The author may need to provide experiments combining VQM, FQM, and PUL to further demonstrate the effectiveness of the method
Questions:
4.   What phenomena might occur if a large number of videos are used to create an open-domain video dataset tethered with closed-domain queries?
Some parts of the article are written in an obscure and difficult-to-understand manner.
5.   "Ours(a), Ours(b), and Ours(c)" in Tables 5, 6, and 7 are mentioned later in subsequent ablation experiments, but there is no explanation in Tables 5, 6, and 7 caption, which may hinder readability.

**Suitability:**

3

---

### Official Review · Reviewer_Roco · 2024-06-06

**Rating:** 2
**Confidence:** 4

**Summary:**

This paper proposes a new setting of Video Moment Retrieval (VMR) called Open-Set Video Moment Retrieval (OS-VMR), where the model should not only localize the temporal boundaries of language queries that are relevant to the video content, but also reject those irrelevant queries describing non-existing visual activities in the video. A new method named OpenVMR is also proposed for the task of OS-VMR. Experiments are conducted to verify the effectiveness of the proposed OpenVMR method.

**Strengths:**

1.The basic motivation of this work is reasonable, i.e., the proposed OS-VMR task is a relatively more practical version of the original VMR task where the model should be capable of recognizing and rejecting irrelevant queries.

2.A relatively comprehensive list of experiments are conducted, e.g., performance comparisons under closed-set and open-set scenarios are reported.

3.The authors give some visualization results of the proposed setting and method, which helps the readers to more intuitively grasp the points of the paper.

**Limitations:**

1.Although the conceptual idea of this paper is basically correct, there are actually considerable flaws in terms of technical soundness and experimental setup. Firstly, this work proposes to learn a distribution from the ID query features and use an OOD boundary to separate the ID queries and OOD queries. However, it is unreasonable to determine OOD queries by merely utilizing the information from the query side, because the relevance between queries and the video can only be correctly acquired based on their semantical matching degree. The OOD query identification is unsolvable if the model has no access to the cross-modal information but only the query information.

2.Secondly, in the open-set experiments of this work, the OOD queries for each video are simply selected from the other datasets. However, there are easily identifiable characteristics among queries from the three datasets in use, i.e., ActivityNet Captions, Charades-STA, and TACoS, since they respectively focus on the open-domain content, indoor activities and cooking scenarios. This provides an obvious but incorrect shorcut for the model to learn and I think this is very likely the reason why the current model performs well on the OOD detection metrics.

3.For the video moment retrieval modules for localizing relevant queries, the designs are short of novelty and lack sufficient insights. Specifically, the video-query matching and frame-query matching module utilize the video-sentence contrastive learning and frame-wise attention gating operations and these are common techniques well investigated in early existing works [1,2,3]. In addition, it seems unclear why the Positive-unlabeled Learning module should adopted for training. Given the reliable ground-truth labels, e.g., the temporal IoU between ID queries and moment proposals, why should we need to convert the training into a semi-supervised learning process that relies on the unreliable cross-modal similarities learned by the model?

4.There are some confusing expressions and results in the paper. For example, in Table 2, Table 3 and Table 4, the type of the proposed model is written as "OS". Combining my personal understanding and the content in Table 1, the closed-set model proposed in this work adopts the same setup as the previous fully-supervised methods, so what additional meaning is expressed by "OS" in the above tables?
Besides, it is a bit confusing to me that the proposed model which is not sophisticatedly designed for the fully-supervised setting turns out to outperform the previous state-of-the-arts by a very significant margin, especially on the ActivityNet Captions and TACoS datasets. Necessary explanations from authors should be added to clarify these confusing points.

Reference

[1] Niluthpol Chowdhury Mithun, Sujoy Paul, Amit K. Roy-Chowdhury. Weakly Supervised Video Moment Retrieval From Text Queries. In CVPR 2019.

[2] Hao Zhang, Aixin Sun, Wei Jing, Joey Tianyi Zhou. Span-based Localizing Network for Natural Language Video Localization. In ACL 2020.

[3] Hao Zhang, Aixin Sun, Wei Jing, Guoshun Nan, Liangli Zhen, Joey Tianyi Zhou, Rick Siow Mong Goh. Video Corpus Moment Retrieval with Contrastive Learning. In SIGIR 2021.

**Suitability:**

3

---

### Meta-Review · Area_Chair_pA6h · 2024-06-28

**Recommendation:** Accept (Oral)
**Confidence:** 5

**Metareview:**

All reviewers have reached an agreement on the acceptance. The paper introduces a novel Open-Set Video Moment Retrieval (OS-VMR) task, addressing the challenge of handling out-of-distribution (OOD) queries in video retrieval. The proposed OpenVMR model effectively distinguishes between ID and OOD queries using normalizing flow technology and demonstrates significant performance improvements over existing methods across multiple datasets. Despite some concerns about methodology and experimental setup, the paper's contributions and experimental validations sufficiently justify its acceptance.